# MAURIVAX: A Vaccination Campaign Project in a Hospital Environment for Patients Affected by Autoimmune Diseases and Adult Primary Immunodeficiencies

**DOI:** 10.3390/vaccines11101579

**Published:** 2023-10-11

**Authors:** Irene Ridolfi, Luca Lo Sardo, Stefania Nicola, Richard Borrelli, Ludovica Comola, Valentina Marmora, Iuliana Badiu, Federica Corradi, Maria Carmen Rita Azzolina, Luisa Brussino

**Affiliations:** 1S.C.D.U. Immunology and Allergology, A.O. Ordine Mauriziano, 10128 Turin, Italy; irene.ridolfi@unito.it (I.R.); llosardo@mauriziano.it (L.L.S.); richard.borrelli@unito.it (R.B.); ludovica.comola@unito.it (L.C.); valentina.marmora@unito.it (V.M.); ibadiu@mauriziano.it (I.B.); fcorradi@mauriziano.it (F.C.);; 2Department of Medical Sciences, University of Turin, 10126 Turin, Italy; 3Health Direction, A.O. Ordine Mauriziano, 10128 Turin, Italy; mazzolina@mauriziano.it

**Keywords:** allergy, hospital vaccination, immunology

## Abstract

**Background:** Patients with autoimmune diseases (ADs) and primary immunodeficiencies (PIDs) are characterized by an increased risk of noninvasive and widespread infections as they are considered frail patients. In addition, many flares of the underlying disease are reported after routine vaccinations. To date, the vaccination rate in these two populations is suboptimal. According to the latest guidelines, targeted interventions are needed, such as strengthening the network of vaccination activities. Our project aimed to propose a pilot network for carrying out the recommended vaccinations in frail patients. **Methods:** The Allergy and Immunology Center of the Mauriziano Hospital in Turin, Italy started the “Maurivax” project, a facilitated pathway for frail patients to administer the recommended vaccinations in the setting of a dedicated structure where they could be properly followed up. **Results:** From June 2022 to February 2023, 49 patients underwent a vaccination consultation: 45 of them (91.8%) were subsequently vaccinated. Among these, 36 subjects (80%) were affected by an active AD and were already in treatment with immunosuppressive therapy or about to start it. Seven patients (15.5%) had a confirmed diagnosis of PID or showed a clinical presentation that was highly suggestive of that condition. Overall, twenty-seven patients (60%) showed a high-grade immunosuppression and six (13.3%) had a low-grade immunosuppression. No patients had a disease flare within 30 days from vaccination and no severe reactions after vaccination was observed. **Conclusions:** Adherence and vaccination safety at our immunology hospital vaccine clinic dedicated to patients with ADs and PIDs were high. We propose an effective model for managing vaccinations in frail patients in a specialist hospital setting.

## 1. Introduction

### Vaccination in Autoimmune Diseases and Primary Immunodeficiencies

Vulnerable or medically frail patients suffer from chronic and complex physical conditions that result in a persistent dependence on specialist nursing and medical care; in the absence of such interventions, they are at risk of irreversible damage and/or death. Among them, patients suffering from oncological diseases, solid organ or hematopoietic stem cell transplantation, severe pulmonary compromise (e.g., severe asthma), autoimmune diseases (ADs), and secondary and primary immunodeficiencies (PIDs) can be listed [1].

ADs include a heterogeneous group of disorders of the immune system, secondary to a loss of tolerance to self-antigens that may be organ-specific (e.g., autoimmune hepatitis, diabetes mellitus type 1a, myasthenia gravis) or systemic [seronegative arthritis, systemic lupus erythematosus (with or without antiphospholipid antibody syndrome), undifferentiated connective tissue disease (UCTD), ANCA-associated vasculitis, rheumatoid arthritis, giant cell arteritis (with or without polymyalgia rheumatica), antiphospholipid antibody syndrome, sjogren’s disease, diffuse systemic sclerosis, sarcoidosis, mixed connective tissue disease (MCTD)] [1].

Patients affected by ADs experience an increased probability of noninvasive and invasive infections due to underlying conditions, comorbidities, and immunosuppressive therapies.

The immunopathogenesis of SLE can be an example of how the disease itself can predispose to infection: type I interferons are in fact persistently produced inducing an alteration of other components of the immune system (e.g., antigen presentation, B cell differentiation, class switching, chemokine production, T cell function). These modulations of the immune system can have an effect on an altered or reduced host defense against pathogens [2].

Immunosuppressive therapies usually aim for remission or a reduction in disease activity in the early stages of the disease, among these there are glucocorticoids, disease-modifying antirheumatic drugs (DMARDs: conventional synthetics—csDMARDs or biologics—bDMARDs), and targeted synthetic DMARDs (tsDMARDs). Conventional synthetic DMARDs include methotrexate, hydroxychloroquine, leflunomide, calcineurin inhibitors, cyclophosphamide, mycophenolate, and azathioprine; -TNFalpha inhibitors, anti-IL6, anti-IL 17, and anti-CD20 drugs are part of the bDMARDs, while jak-inhibitors belong to the tsDMARDs.

The PIDs include several genetic anomalies that affect different components of the innate and adaptive responses that lead to an increase in the incidence, frequency, or severity of infections and/or immune dysregulation. Defects in the adaptive arm of the immune system include combined immunodeficiencies and antibody deficiency syndromes; abnormalities in innate immunity are defects of phagocytes, the complement pathway, or toll-like receptor-mediated signaling.

As with ADs, patients with PIDs are more susceptible to even serious and invasive infections and are more often exposed to pathogens by attending healthcare facilities more often when compared with the general population. The pathogen that causes the infection in a patient with immunodeficiency depends on the type of immune deficiency. Antibody deficiencies are associated with infections with Gram-positive bacteria; cellular deficiencies with infections by mycobacteria, protozoa, fungi, viruses, and opportunistic bacteria; and phagocytic disorders with infections by staphylococcal, fungal, and gram-negative organisms [3].

Therefore, the prevention of complicated infections is crucial and recommended in the management of these patients. Vaccinations can prevent some diseases by inducing and/or strengthening the immune system.

In patients affected by PIDs, all the vaccines foreseen by the vaccination schedule can be performed when effective and safe. They can receive all recombinant vaccines consisting of the killed/inactivated germ. Protective conjugate vaccines against capsulated germ infections (pneumococcus, meningococcus—strains A, C, Y, W135, and B—and Haemophilus influenzae) are essential, especially in antibodies and complement deficiencies. Live virus vaccines (measles, rubella, mumps, chicken pox, yellow fever, tuberculosis, herpes zoster, oral poliomyelitis, oral typhoid fever) are contraindicated for some types of PIDs, where T cells are missing or malfunctioning.

In PIDs, vaccine response may not be as effective because it depends on the underlying immunological defect.

ADs’ inactivated vaccines (i.e., *influenza*, *S. pneumoniae*, *hepatitis A*, *hepatitis B*, *human papillomavirus*, *toxoid tetanus*, *H. influenzae b*., *N. meningitidis*, *diphtheria*, *pertussis*, *parenteral poliomyelitis*, *encephalitis* transmitted by ticks, parenteral typhoid fever) can be administered in accordance with the recommendations for the general population, preferably 2 weeks or more before the start of immunosuppressive therapy. Live attenuated vaccines should be avoided or performed at least 2–4 weeks before starting immunosuppressive therapy or 3 months after discontinuation (for rituximab, 4 weeks before the start of therapy or at least 12 months after the last infusion) [1].

To date, however, the vaccination rate in both ADs and PIDs is suboptimal; it may depend on the low request for vaccination by the health practitioners (such as rheumatologists, immunologists, and general practitioners) who are taking care of the patients, the lack of trust in the effectiveness of vaccination, especially from the patients, or the variable and individual degree of immunological impairment and consequent vaccine response [1].

According to the latest recommendations for the vaccination of adult patients with ADs [1] and PIDs [3], the Italian National Vaccine Prevention Plan (PNPV) 2023–2025, and European [4] and global [5] guidelines, the prevention of infectious diseases through vaccination in these patients is a Public Health priority and requires targeted interventions including the strengthening of the vaccination activity network for mandatory and recommended vaccinations.

In Piedmont, vulnerable adults affected by ADs or PIDs are currently vaccinated in vaccination clinics managed by the territorial local health authorities (A.S.L.). Excluding vaccination for SARS-CoV2, there are only a few permanent vaccination realities in hospitals offered to patients at risk in Italy, particularly in four regions: Lombardy, Liguria, Emilia-Romagna, and Trentino-Alto Adige [6].

The Allergy and Immunology Unit of Mauriziano Hospital in Turin, Italy has been actively involved in the vaccination campaign for SARS-CoV-2 in fragile patients suffering from PIDs and systemic ADs; for these patients, the service has set the goal to improve the involvement for vaccinations, facilitating both the access and the administration of the recommended vaccinations for these patients with a service performed by the Immunologists who were taking care of the patients. The name “Maurivax” (a combination of “Mauriziano” and “Vaccination”) was chosen for the project. Therefore, the objective of our study was to create easier access to vaccination in patients with ADs and PIDs and to evaluate vaccine adherence at our specialized vaccination hospital clinic.

## 2. Methods

A 12 month-period was scheduled for the project (March 2022–February 2023). An Immunologist was selected to perform this service with a dedicated structure and agenda in order to perform vaccinations once a week; all the necessary resources required for vaccinations (work station, medical bed, stretcher, emergency trolley, refrigerator, vaccine products available, materials for vaccination practice such as syringes, disinfectants, cotton, plasters, bio boxes) were made available for this purpose.

Thanks to the collaboration of the A.S.L. Città di Torino (Turin, Italy) to start the project, at our request an initial quantity of capsulated bacterial vaccines and the new recombinant herpes zoster vaccine were supplied. Therefore, the following non-live vaccines were made available: the 13-valent conjugate anti-pneumococcal vaccine (PCV13), the 23-valent polysaccharide anti-pneumococcal vaccine (PPSV23), the tetanus toxoid conjugate vaccine for H. influenzae, the recombinant DNA vaccine for N. meningitidis B, the conjugate vaccine for N. meningitidis serotypes A, B, C, Y, W-13, and the recombinant, adjuvanted vaccine for the H. zoster virus.

The project was divided into two phases. The initial phase consisted of immunization and vaccination counseling of patients, characterized by assessment of vaccination status [access and consultation of the “Regional Information System for Vaccination Management” (SIRVa) regional online platform], indications for further vaccinations (vaccination history, adverse events in previous vaccinations), disease activity status up to 3 months before vaccination, and a therapy current or planned immunosuppressive disorder; in this phase, vaccinations were scheduled and cohabiting family members and caregivers were encouraged to get vaccinated according to national guidelines (except the oral polio vaccine). In the second phase, the patients were subjected to suitable counseling and, after signing the informed consent, were vaccinated according to the vaccination schedule for each vaccine and the precautions for each pathology indicated by the main guidelines. Table 1. shows a summary of the principles and dosages adopted for each vaccine available during the project [1,3,7,8,9].

Consecutive patients over 18 years old, suffering from ADs and/or suffering or highly suspected of PIDs, belonging to our Immunology and Allergology department who had agreed to participate in the study were included in the study. Patients who did not provide informed consent were excluded.

During the project, demographic data (age, gender), clinical data (pathological and drug history, risk factors for serious and/or invasive infectious diseases, such as transplantation, splenectomy, severe asthma, oncological disease), and vaccination practice (previous infections, vaccines and adverse effects, vaccines performed, and related adverse effects) were collected for each patient in a prepared database.

Each vaccinated patient had read and understood the information sheet and signed a written informed consent before vaccination and data collection.

The study was submitted to a corporate ethics committee (File No. 266/2022; Prot. N° 0085256–2/8/2022; Protocol code: MAURIVAX; No.: 644.423; backend number: 00266/2022), conducted in full compliance with the latest revision of the Declaration of Helsinki and according to the latest Best Practice guidelines.

Non-live vaccines were administered in patients with ADs receiving immunosuppressive therapy; in those requiring it, vaccination was performed 2 weeks or more after initiation [11]; for rituximab, vaccination was provided at least 6 months after administration and 4 weeks before the next course of B-cell depleting therapy [1].

Inactivated vaccines have been administered in PID, regardless of the timing of the replacement immunoglobulin infusion [11].

Statistical analysis was performed using IBM SPSS Statistics for Windows software, version 28 (IBM Corp., Armonk, NY, USA). A descriptive analysis of the studied variables was conducted. The normality of the distribution of continuous variables was evaluated with the Shapiro–Wilk and the Kolmogorov–Smirnov test. The distribution of the qualitative variables in absolute number and percentage has been reported and the relative frequency tables have been created. For the quantitative variables, based on the normality of the distribution, mean, standard deviation, or median, interquartile ranges were calculated. The results were considered significant for p-values less than 0.05.

### Stratification of the Degree of Immunocompromise

Patients were stratified according to the degree of immunocompromise as per the Center for Disease Control and Prevention (CDC) [12] and the Italian Medicines Agency (AIFA) [13] criteria. Subjects with high levels of immunosuppression were selected along with those with severe combined immunodeficiency—SCID, common variable immunodeficiency—CVID, DiGeorge syndrome, and Wiskott–Aldrich syndrome, high-dose corticosteroid maintenance recipients (PDN ≥ 20 mg/day, for ≥14 consecutive days), and recipients of biological immunomodulators (e.g., interleukin inhibitors, calcineurin inhibitors, TNF alpha inhibitors, JAK inhibitors, selective immunosuppressants, anti-CD-20, alkylators, antimetabolites); in this group, patients receiving combined therapies regardless of the dose taken were included as well.

On the other hand, subjects with low/moderate immunosuppression who received short-term (<14 days) systemic corticosteroids, long-term corticosteroids with short-acting preparations, corticosteroids as maintenance replacement therapy, doses of prednisone equivalents ≤ 20 mg/day for ≥14 consecutive days, or were receiving methotrexate (MTX) <0.4 mg/kg/week or azathioprine <3mg/kg/day were also considered. All other possible therapies taken in monotherapy or combination were considered a high degree of immunosuppression.

## 3. Results

From June 2022 to February 2023, we carried out 49 vaccination counseling sessions: 45 of these patients were subsequently vaccinated (91.8%) at our Service and 114 total doses were administered (doses stratified by degree of immunosuppression available in the Appendix A).

Among the forty-five vaccinated patients, thirty-six (80%) were affected by ADs (Table 2) and on or about to start immunosuppressive therapy, seven (15.5%) had confirmed or highly suggestive PID, one (2.2%) had received an organ transplant (liver) on calcineurin inhibitor therapy, and one patient had severe asthma (2.2%) on chronic methylprednisolone therapy (prednisone equivalents < 20 mg/day).

The mean age of the series was 59.6 [range: 21–89] years and females were 30 (66.6%).

Seventeen patients (34.6%) were affected by controlled arterial hypertension and twenty-one (45.6) by gastroesophageal reflux. The majority of patients had never smoked (63.2%) and 18 patients smoked or were former smokers (36.7%).

### 3.1. Total Case Series: PID and ADs

Among the forty-five vaccinated patients, twenty-seven (60%) had confirmed high-grade immunosuppression and six (13, 3%) had confirmed low-grade immunosuppression.

#### 3.1.1. Autoimmune Diseases

Of the thirty-six vaccinated patients suffering from ADs, at the time of vaccination, twenty-six (57.7%) were on immunosuppressive therapy: sixteen (65.3%) on immunosuppressive combination therapy (high-grade immunosuppressants), eight on a single immunosuppressive drug of high grade, and two on a single low/moderate grade immunosuppressive drug (Table 3 and Table 4 shows the distribution of patients taking each therapy). Notably, only one patient had undergone vaccination 8 months after rituximab infusion. Of the remaining nine patients, three were not taking immunosuppressive drugs, four (11.1%) were taking hydroxychloroquine, one patient had systemic sclerosis on scheduled iloprost infusions, and one patient was scheduled to start immunosuppressive therapy.

No patient in the series with the ADs had a disease recurrence/exacerbation within 30 days following vaccination. In one case, the second scheduled vaccine dose for recombinant herpes zoster was delayed to 6 months due to the finding of active disease (psoriatic lichenoid papular eruption during cyclosporine).

#### 3.1.2. Primary Immunodeficiencies (PID)

Table 5 shows the vaccinated patients affected by PID: two patients had symptoms highly suggestive of common variable immunodeficiency in the course of diagnostic definition. The five vaccinated patients with a confirmed diagnosis of PID were receiving replacement doses of intravenous immunoglobulin (IVIG).

#### 3.1.3. Administered Vaccines and Adverse Effects

To date, forty-two (93.3%) patients have been vaccinated with both two doses of adjuvanted recombinant herpes zoster vaccine (Shingrix—GSK); eight of these (19%) had at least one episode of herpes zoster in their lifetime and three (7.1%) had one episode in the previous 12 months; among the patients vaccinated for herpes zoster, ten (23.8%) reported pain at the injection site, three (7.1%) had an extensive local reaction with spontaneous resolution within 24 h, two (4.7%) experienced asthenia in the next 24 h, and four (9.5%) experienced a low-grade fever (maximum temperature: 38 °C) in the first 24 h after dosing resolved in up to 2 days; one patient (2.3%) reported choking sensation and hoarseness 1 min after dosing, promptly treated with systemic corticosteroids and antihistamines with a resolution of symptoms (this patient was suffering from asthma and antibody deficiency and had previously reported similar reactions with the meningococcal C vaccine; he was subsequently given a hepatitis B vaccine by us without complications).

Additionally, at the second dose of the Shingrix vaccine, one patient (2.3%) experienced an extensive local reaction that resolved within a week and one patient experienced recurrent arthralgias (diagnostic workup currently in progress).

Four patients (8.8%) underwent the group A, C, W-135, and Y meningococcal conjugate vaccine (Menveo—GSK), five (11.1%) the group B recombinant meningococcal DNA vaccine (Bexsero—GSK), seven (15.5%) the 13-valent pneumococcal conjugate vaccine (PCV13, Prevnar-13 -Pfizer), and two (4.44%) the 23-valent pneumococcal polysaccharide vaccine (PPSV23, Pneumovax -MSD).

## 4. Discussion

We presented the first adherence data of a hospital vaccination clinic dedicated to patients with AD and PID activated at the Univerisity Unit of Immunology and Allergology of the Mauriziano Hospital in Turin, Italy.

The data show that a larger number of patients with AD have been vaccinated when compared with those affected by PID: this may be related to a known higher prevalence of ADs in the general population (3–5%) [14] compared with the prevalence of PID (this prevalence varies according to the type of immunodeficiency: new PIDs are discovered continuously and are more frequently diagnosed; however, the exact and overall prevalence is hitherto not known, but estimated to be low) [15].

Most vaccinated cases were women compared with men, which could be related to the higher prevalence of ADs among vaccinated subjects [16].

The most frequent ADs in patients who have been vaccinated were seronegative spondyloarthritis (ankylosing spondylitis, psoriatic arthritis, reactive arthritis, and enteropathic arthritis), systemic lupus erythematosus, and undifferentiated connective tissue disease: these data are in line with the prevalence of the diseases in Europe [17,18,19].

In our study, we underline the adherence and safety data for the administered vaccines and propose a model of a specialized hospital immunological vaccination clinic.

Thanks to the dedicated service we managed to create for these patients at risk, we have achieved a level of adherence to vaccinations after proper counseling in over 90% of cases; this rate, for the same vaccines performed, is above the rates of adherence to vaccines reported globally for cases of patients with ADs [20,21] or with other types of diseases at risk (diabetics [22,23], chronic lung diseases [24], splenectomized [25] patients).

The vaccination counseling performed by the Immunologist allowed for creating individualized vaccination plans (indication for further vaccinations) based on the disease (activity status, severity, and current or planned therapy). Such scenarios allowed to perform vaccinations during the quiescent phase of the disease and before starting immunosuppressive therapy (especially when it comes to B-cell depleting therapy).

We highlight that this is reflected in the absence of significant exacerbations of the ADs after the administration of vaccines and the absence of serious side effects in the majority of patients; the only patient who experienced a serious reaction was promptly treated with a complete resolution of the symptoms: this was guaranteed and obtained thanks to the organization of the clinic and the presence of a trained healthcare team, but also the allergy knowledge of the specialist immunologist responsible for these patients.

Our Maurivax project will allow us to periodically measure vaccination coverage (we will propose an annual assessment of vaccination status) and to promote vaccination to all eligible patients during routine immunological visits, but concomitantly also to cohabitants, caregivers, or family members. The immunologist can explain the characteristics of the vaccine to be performed directly to the patient in charge of the facility and to their cohabitants (adverse effects, efficacy), obtaining a shared and early decision-making process to increase vaccination adherence.

We recognized some limitations in our study. According to the stratification proposed by us, more than half of the patients subjected to vaccination presented a HIGH GRADE (primary or secondary) immunosuppression. To date, the immunogenicity and the preventive effect obtained from the vaccinations carried out are not known; however, this is currently under evaluation. In addition, to start the project we decided to request vaccines for capsulated bacteria and the new recombinant vaccine for herpes zoster; however, one of our goals will be to increase the availability of the range of recommendable vaccines in these populations.

Another goal of ours will be to simplify access to our immunological vaccination center for patients with primary or secondary immunosuppression from other hospital departments; this could be ensured by the implementation of an interdepartmental and/or intercompany protocol and by the creation of a medical service that allows for electronic patient booking.

Furthermore, it will be necessary to expand our cohort of vaccinated patients: in light of the complexity and variability of autoimmune diseases, a larger sample size could improve the generalizability of our results.

With the results shown, we hope to encourage the creation of other specialized immunological vaccination clinics in Italy, similar to the Maurivax project, as they allow us to obtain numerous other advantages such as the reduction in direct and indirect costs of the disease by preventing complicated and invasive infections, the reduction in patients’ travel and absences from work to carry out vaccinations in other health facilities, and the increase in vaccine monitoring information for the scientific community in the population of patients with ADs and PIDs, contributing to counteracting the antibiotic phenomenon of resistance in immunosuppressed subjects who have often already undergone cycles of antibiotic therapy at higher doses and spectra.

## 5. Conclusions

For patients with ADs and PIDs, specialized hospital clinics can represent an effective public health strategy to ensure vaccination adherence in these fragile patients and their caregivers, identifying the correct vaccines and administration times. The results of our study support excellent vaccination adherence; therefore, we suggest that the proposed model may be exportable to other immunology and rheumatology clinics in Italy.

## 6. Synthesis

Our Maurivax project can be an exportable model of in-hospital specialist vaccination to increase both adherence and safety to vaccination in patients with autoimmune diseases and primary immunodeficiencies.

## Figures and Tables

**Table 1 vaccines-11-01579-t001:** Vaccination principles and dosages for each type of vaccine available adopted during the Maurivax project.

Vaccines Available in the Maurivax Project	Indications of the Vaccine in Autoimmune Diseases or Primary Immunodeficiencies According to Guidelines and Product Technical Sheet
Haemophilus influenza serotype B (HiB) tetanus toxoid conjugate vaccine (Hiberix—GSK)	- From data sheet: number of doses based on age; single dose in adults; - Especially recommended in [8] familial complement deficiencies (e.g., C3 or C5 deficiencies) or treatments that inhibit terminal complement activation (e.g., eculizumab), anatomical or functional asplenia or subjects awaiting elective splenectomy, congenital or acquired immunodeficiencies (such as lack of antibodies especially for the IgG2 subclass or HIV-positive subjects, bone marrow transplant recipients or awaiting a solid organ transplant, individuals undergoing chemotherapy or radiation therapy for the treatment of malignancies, recipients of cochlear implants
Streptococcus pneumoniae 13-valent conjugate vaccine (PCV13) and 23-valent polysaccharide vaccine (PPSV23)	- Sequence according to CDC and European Society of Clinical Microbiology and Infectious Diseases (ESCMID) recommendations [10]; single dose in adults; - Recommended for those with predisposing diseases or conditions such as chronic heart disease, chronic lung disease, diabetes mellitus, chronic liver disease (including cirrhosis of the liver and alcohol-induced chronic liver disease), chronic alcoholism, individuals with CSF leaks from trauma or surgery, presence of a cochlear implant, hemoglobinopathies (sickle cell anemia and thalassemia), congenital or acquired immunodeficiencies, HIV infection, conditions of anatomical or functional asplenia and patients candidates for splenectomy, onco-hematological diseases (leukemia, lymphoma, and multiple myeloma), widespread neoplasms, organs or bone marrow, diseases requiring long-term immunosuppressive treatment, chronic renal/adrenal insufficiency [8].
Neisseria meningitides Serogroups A, C, W135, and Y conjugate vaccine (Menveo—GSK)	- Number of doses from the data sheet according to age [3]; single dose in adults; - Recommended in patients with hemoglobinopathies such as thalassemia and sickle cell anemia, functional or anatomical asplenia, and candidates for elective splenectomy, congenital or acquired immunosuppression (particularly in case of organ transplantation, antineoplastic, or systemic therapy with high doses of corticosteroids), diabetes mellitus of type 1, chronic renal/adrenal insufficiency, HIV infection, severe chronic liver disease, CSF leak from trauma or surgery, congenital complement defects (C5—C9), Tolls such as type 4 receptor and properdin defects [8].
Herpes Zoster Virus (HZV) recombinant vaccine, adjuvanted (Shingrix—GSK)	- For adults 18 years of age and older if increased risk of HZ *; - For individuals who are or may become immunodeficient or immunosuppressed due to disease or medication and who would benefit from a shorter vaccination schedule; the second dose may be given 1 to 2 months after the initial dose *; - Offered in the presence of diabetes mellitus, cardiovascular pathology, COPD, or subjects destined for immunosuppressive therapy [8].
Neisseria meningitidis serogroup B (MenB) recombinant DNA vaccine (Bexsero—GSK)	- Number of doses from the data sheet according to age [3]; two doses in adults; - Recommended in patients with hemoglobinopathies such as thalassemia and sickle cell anemia, functional or anatomical asplenia, and candidates for elective splenectomy, congenital or acquired immunosuppression (particularly in case of organ transplantation, antineoplastic or systemic therapy with high doses of corticosteroids), diabetes mellitus of type 1, chronic renal/adrenal insufficiency, HIV infection, severe chronic liver disease, CSF leak from trauma or surgery, congenital complement defects (C5–C9), Tolls such as type 4 receptor and properdin defects [8].

*: technical data sheet of the drug.

**Table 2 vaccines-11-01579-t002:** Disease distribution of vaccinated patients with ADs.

Autoimmune Diseases	Distribution (N° Patients—%)
Seronegative arthritis	7 (15%)
Systemic Lupus Erythematosus (with or without Antiphospholipid Antibody Syndrome)	6 (13, 3%)
Undifferentiated connective tissue disease (UCTD)	4 (8, 8%)
ANCA-associated vasculitis	3 (6, 6%)
Rheumatoid arthritis	2 (4, 4%)
Giant cell arteritis (with or without polymyalgia rheumatica)	2 (4, 4%)
Antiphospholipid Antibody Syndrome	1 (2, 2%)
Sjogren’s disease	1 (2, 2%)
Diffuse systemic sclerosis	1 (2, 2%)
Sarcoidosis	1 (2, 2%)
Polymyalgia Rheumatica	1 (2, 2%)
Mixed connective tissue disease (MCTD)	1 (2, 2%)
Psoriasis	1 (2, 2%)
Prurigo nodularis	1 (2, 2%)
C3 nephropathy	1 (2, 2%)
Localized scleroderma	1 (2, 2%)
Vogt-Koyanagi Harada syndrome	1 (2, 2%)
Autoimmune polyendocrinopathy type 1 or APECED syndrome	1 (2, 2%)

**Table 3 vaccines-11-01579-t003:** Disease distribution of vaccinated patients with AD, divided by severity of immunosuppression.

Autoimmune Disease-Vaccinated Patients	Type of Immunosuppressive Therapy	N° (%) Patients	Distribution of Therapy—N°
**Not taking immunosuppressive therapy**		9 (25%)	No maintenance therapy 3 (8, 3%) Hydroxychloroquine 4 (11, 1%) Iloprost infusion 1 (2, 7%) Waiting to start immunosuppressive therapy 1 (2, 7%)
**Taking Immunosuppressive therapy**	**Combined high grade**	16 (44, 4%)	GC + CYS 1 (2, 7%) GC + MMF 3 (8, 3%) GC + JAKi 1 (2, 7%) GC + MTX 2 (5, 5%) GC + MTX + JAKi 1 (2, 7%) GC + anti-IL 17 1 (2, 7%) GC + AZA 2 (5, 5%) GC + CYF 1 (2, 7%) MTX + anti-IL 17 1 (2, 7%) MTX + ADA 1 (2, 7%) MTX + JAKi 1 (2, 7%) AZA + ADA 1 (2, 7%)
	**Single high grade**	8 (22, 2%)	MMF 2 (5, 5%) MTX 2 (5, 5%) GC 1 (2, 7%%) CALi 1 (2, 7%%) IL6Ri 1 (2, 7%%) Rituximab 1 (2, 7%)
	**Single low/moderate grade**	3 (8, 3%)	GC 2 (5, 5%) MTX 1 (2, 7%%)

**Table 4 vaccines-11-01579-t004:** Distribution by therapy of vaccinated patients with ADs on high-grade immunosuppressive therapy.

Immunosuppressive Therapy	N°(%) Vaccinated Patients with ADs on High-Grade Immunosuppressive Therapy (Single or Combination)
Glucocorticoids *	14 (53, 8%)
Methotrexate *	9 (34, 6%)
Mycophenolate mofetil	5 (19, 2%)
Azathioprine *	3 (8, 3%)
JAK inhibitors	3 (8, 3%)
IL17A inhibitors	2 (7, 6%)
Anti-TNF alpha	2 (7, 6%)
Calcineurin inhibitors	2 (7, 6%)
IL6R inhibitor	1 (2, 7%)
Anti-CD20	1 (2, 7%)
Cyclophosphamide	1 (2,7%)

*: high-grade immunosuppressive dosages for corticosteroids, methotrexate, and azathioprine when taken individually); CALi: calcineurin inhibitors; GC: glucocorticoids; CYS: cyclosporine; MMF: mycophenolate mofetil; JAKi: JAK inhibitors; AZA: azathioprine; IL17i: IL17A inhibitors; IL6iR: IL6 receptor inhibitor; ADA: adalimumab.

**Table 5 vaccines-11-01579-t005:** Distribution of vaccinated patients with confirmed or highly suspected PID.

PID (Confirmed or Highly Suggestive)	Distribution (N° Patients—%)	IVIG Therapy
Highly suspected common variable immunodeficiency being diagnosed	2 (28, 5%)	no
Confirmed Common Variable Immunodeficiency (CVID)	2 (28, 5%)	yes
Confirmed IgG deficiency	2 (28, 5%)	yes
Confirmed X-linked agammaglobulinemia	1 (14, 2%)	yes

## Data Availability

The data that support the findings of this study are available from the corresponding author, [N.S.], upon reasonable request.

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
