# Peer review of "MAURIVAX: A Vaccination Campaign Project in a Hospital Environment for Patients Affected by Autoimmune Diseases and Adult Primary Immunodeficiencies"

_vaccines, 2023, doi:10.3390/vaccines11101579_

Round 1

Reviewer 1 Report

The topic of vaccination of fragile patients in hospital by their specialists is very prevailing, therefore the paper can be published, but requires minor revisions:

In the "introduction" section on lines 75-82, when defining the timing of vaccinations with respect to the timing of therapy administration, references are necessary (guidelines if they exist).

At the end of the introduction the objective of the study must be better defined, considering that at the beginning of the "discussion" section (line 244) it is stated that the efficacy data of vaccination in hospital are presented.

In the "methods" section (line 120) reference is made to the assessment of vaccination status: how was it assessed? (self reported?).

In the "results" section (lines 181-182) it is stated that 45 patients were vaccinated and that 81 doses were administered. Of which vaccines per category of patients stratified by degree of immunocompromise as described in the "methods" section? (lines 161-178).

The "discussion" section begins by saying that the first data on the efficacy of vaccination in hospital are presented: but how was the efficacy measured? On lines 261-262 it is said that the levels of adherence to vaccination changed after counseling in 90% of cases: by how much did this adherence change and compared to which starting values?

On line 288 it says that only a few vaccines were provided to start the project: which ones and why?

Author Response

Reply to reviewer 1

Dear reviewer,

we thank you for your availability and for the suggestions received. Below we list the Comments and Suggestions for Authors and the respective responses:

  1. In the "introduction" section on lines 75-82, when defining the timing of vaccinations with respect to the timing of therapy administration, references are necessary (guidelines if they exist).

In the revised version, many new bibliographic references have been added.

  1. At the end of the introduction the objective of the study must be better defined, considering that at the beginning of the "discussion" section (line 244) it is stated that the efficacy data of vaccination in hospital are presented.

At the end of the introduction (lines 123-125), a sentence was added to define the objective of the study.

  1. In the "methods" section (line 120) reference is made to the assessment of vaccination status: how was it assessed? (self reported?).

In the "methods" section (lines 143-144), the method with which we were able to evaluate the vaccination status of the patients was added (not with the patient's self-declaration, but with a healt-care driven regional online platform).

  1. In the "results" section (lines 181-182) it is stated that 45 patients were vaccinated and that 81 doses were administered. Of which vaccines per category of patients stratified by degree of immunocompromise as described in the "methods" section? (lines 161-178).

As requested, we created a table with the number of doses administered for each vaccine, stratifying them by degree of immunosuppression. We have recalculated the number of doses actually administered: they are 114 and not 81, so we have modified this sum.

We propose to include the following table and its caption as supplementary material, so we have included the citation of the supplementary material on line 207-208 of the manuscript.

Autoimmune disease-vaccinated patients

Type of immunosuppressive therapy

N° (%) patients

N°of administered doses

Shingrix

Prevenar 13

Pneumovax

Bexsero

Menveo

Hiberix

Not taking immunosuppressive therapy

9 (25%)

18

2

1

2

1

0

Taking Immunosuppressive therapy

Combined high grade

16 (44,4%)

32

2

0

2

1

0

Single high grade

8 (22,2%)

16

0

0

2

1

0

Single low/moderate grade

3 (8,3%)

6

1

0

2

1

1

PID (confirmed or highly suggestive)

N° (%) patients

Shingrix

Prevnar-13

Pneumovax

Baxero

Menveo

Hiberix

Highly suspected common variable immunodeficiency being diagnosed

2 (28,5%)

4

0

0

0

0

0

Confirmed Common Variable Immunodeficiency (CVID)

2 (28,5%)

4

1

1

0

0

0

Confirmed IgG deficiency

2 (28,5%)

4

0

0

2

0

0

Confirmed X-linked agammaglobulinemia

1 (14,2%)

2

0

0

0

0

0

Organ transplant (liver)

1

2

1

0

0

0

0

Severe asthma

1

2

0

0

0

0

Total

45

90

7

2

10

4

1

Table S1. Doses stratified by degree of immunosuppression and disease.

  1. The "discussion" section begins by saying that the first data on the efficacy of vaccination in hospital are presented: but how was the efficacy measured? On lines 261-262 it is said that the levels of adherence to vaccination changed after counseling in 90% of cases: by how much did this adherence change and compared to which starting values?

On line 270 we changed the wording "efficacy data" to "adherence data" as our study had the primary objective of evaluating adherence to our new project and model of intra-hospital vaccination.

We do not have vaccination adherence data before the implementation of the Maurivax project, therefore we cannot indicate the percentage change. We can only say that among the people subjected to our vaccination counseling, up to 90% of these patients have joined the project. On line 267 we therefore changed the term "change of adhesion" to the term "achieving a level of adhesion".

  1. On line 288 it says that only a few vaccines were provided to start the project: which ones and why?

In lines 133-135 we specified that the supply of vaccines for the start of the project was initially requested by us only for recombinant herpes zoster as it was recently on the market and for the capsulated bacterium which we considered a priority on fragile patients suffering from AD and PID.

On lines 314 and 315 we specified the project began at our request with the vaccination for capsulated bacteria and for the new recombinant vaccine for herpes zoster, but that we intend to extend it to the remaining vaccinations recommended in these categories of patients.

Reviewer 2 Report

Ridolfi et al. submitted the manuscript titled: "MAURIVAX: a vaccination campaign project in a hospital environment for patients affected by autoimmune diseases and 3 adult primary immunodeficiencies". It represents the Maurivax" project, a facilitated pathway for frail patients to administer the recommended vaccinations in the setting of a dedicated structure where they could be properly followed up.

The manuscript is well-written and well-structured overall. It may, however, benefit from some clarity and flow enhancements. For example, the introduction might include a more in-depth summary of autoimmune disorders, adult main immunodeficiencies, and their vaccine consequences. Furthermore, a clear definition of the research aims and hypotheses driving the investigation would be beneficial.

The authors note the need for a bigger cohort, which is an important aspect. Given the complexities and variability of autoimmune illnesses, a larger sample size might improve the study's generalizability.

The authors briefly address gender in the development of autoimmune illnesses but do not go into detail about it. A gender-specific investigation would be useful to determine whether females are truly more impacted by AID than males. If such differences are discovered, potential underlying variables or mechanisms should be investigated.

Information on inclusion/exclusion criteria and their justification would be useful. It would also be useful to know whether patient demographics were taken into account throughout the selecting process.

The MAURIVAX immunization program is unquestionably a worthwhile initiative in meeting the vaccine needs of immunocompromised people. However, minor modifications are required to guarantee scientific integrity, clarity, and a more thorough investigation of gender differences in autoimmune disorders. Once these challenges are resolved, the study has the potential to significantly contribute to the area of vaccination in vulnerable groups.

Author Response

Dear reviewer,

we thank you for your availability and for the suggestions you gave us. Below, we list the Comments and Suggestions for Authors and the respective responses:

  1. The introduction might include a more in-depth summary of autoimmune disorders, adult main immunodeficiencies, and their vaccine consequences.
  1. in lines 42-48, we mentioned some autoimmune diseases differentiated between organ-specific and systemic. In lines 52-57, we have described an example of the mechanism by which AD pathology can predispose to infection and therefore to the need for vaccination.
  2. we have added lines from 74 to 79: an in-depth analysis of the most common infections in relation to the type of immunodeficiency.
  1. A clear definition of the research aims and hypotheses driving the investigation would be beneficial.
  1. the hypotheses driving the study are reported on lines 101-116. Here, we highlighted the lack of vaccination in fragile patients suffering from ADs and PIDs,. Thus, we described the current situation in the Piedmont region and the role that a specialist hospital structure could have, as proposed by our project.
  2. the objective of the study has been explored in detail in lines 123-125.
  1. The authors note the need for a bigger cohort, which is an important aspect. Given the complexities and variability of autoimmune illnesses, a larger sample size might improve the study's generalizability.
  1. Thanks for the observation. We added this study limitation in lines 323-325. We are aware that the project is a pilot project and can be improved in the next years.
  1. The authors briefly address gender in the development of autoimmune illnesses but do not go into detail about it. A gender-specific investigation would be useful to determine whether females are truly more impacted by AID than males. If such differences are discovered, potential underlying variables or mechanisms should be investigated.
  1. Thank you for your observation. At the moment we have only described the higher prevalence of women among vaccinated patients with AD. A greater prevalence of the female gender among patients with ADs is currently already widely described in the literature. The pathogenesis and mechanisms by which this occurs fell outside the aim of this study.
  1. Information on inclusion/exclusion criteria and their justification would be useful.
  1. the inclusion and exclusion criteria of the study were listed in lines 157-160
  1. It would also be useful to know whether patient demographics were taken into account throughout the selecting process.
  1. in line 157, among the inclusion criteria, an age greater than 18 years was specified among the demographic criteria.